# Carbon Fiber Reinforced Multi-Phase Epoxy Syntactic Foam (CFR-Epoxy-Hardener/HGMS/Aerogel-R-Hollow Epoxy Macrosphere(AR-HEMS))

**DOI:** 10.3390/polym13050683

**Published:** 2021-02-24

**Authors:** Xinfeng Wu, Yuan Gao, Tao Jiang, Ying Wang, Ke Yang, Tengshi Liu, Kai Sun, Yuantao Zhao, Wenge Li, Jinhong Yu

**Affiliations:** 1College of Ocean Science and Engineering and Merchant Marine College, Shanghai Maritime University, Shanghai 201306, China; 201830410075@stu.shmtu.edu.cn (Y.G.); jiangtao9585@163.com (T.J.); 201940110009@stu.shmtu.edu.cn (Y.W.); Kais@shmtu.edu.cn (K.S.); zhaoyt@shmtu.edu.cn (Y.Z.); wgli@shmtu.edu.cn (W.L.); 2School of Materials Science and Engineering, Central South University, Changsha 410083, China; 3School of Materials Science and Engineering, Shanghai University, Shanghai 200444, China; liutengshi@shu.edu.cn; 4Key Laboratory of Marine Materials and Related Technologies, Zhejiang Key Laboratory of Marine Materials and Protective Technologies, Ningbo Institute of Materials Technology & Engineering, Chinese Academy of Sciences, Ningbo 315201, China

**Keywords:** polymer composites, epoxy syntactic foam, aerogel, carbon fiber, density, compressive strength

## Abstract

Because the aerogel has ultra-low density and good impact resistance, the aerogel material, epoxy-hardener system, and expandable polystyrene beads (EPS) were used to prepare the lightweight aerogel reinforced hollow epoxy macro-spheres (AR-HEMS). The multi-phase epoxy syntactic foam (ESF) was manufactured with the epoxy-hardener system, HGMS (EP-hardener-HGMS), and AR-HEMS by “the compression modeling method.” In this experiment, in order to enhance the strength of the ESF, some different kinds of the carbon fiber (CF) were added into the EP-hardener-HGMS system (CFR-EP). The influence of the volume stacking fraction, inner diameter, and layer of the AR-HEMS and the content and type of the CF in the EP-HGMS (CFR-EP) system on the compressive strength of the ESF were studied. Weighing the two factors of the density and compressive strength, the ESF reinforced by 1.5 wt% CF with 90% AR-HEMS has the better performance. This kind of the ESF has 0.428 g/cm^3^ nd 20.76 Mpa, which could be applied in 2076 m deep sea.

## 1. Introduction

As a white powder material with extremely high porosity, aerogel is widely used in the fields of building insulation materials and military bulletproof materials due to its excellent thermal insulation resistance and thermal energy storage [1]. As one of the lowest density solid materials in the world, its porosity can reach 99%, and its density can reach 0.13~0.53 mg/cm^3^. Because of their low weight, aerogel materials are favored in the fields of aerospace and deep-sea exploration [2]. In recent years, the exploration of the deep-sea field and resource exploration activities have become more frequent, and the development of our country in deep-sea exploration are in the forefront of the world. In June 2019, the ‘Jiaolong’ probe dive reached the bottom of the sea at 7062 m, breaking the world record for diving with similar operating subsea equipment. Lightweight buoyancy materials have broad application prospects in such deep-sea exploration equipment [3,4]. Deep-sea buoyancy materials have undergone a period of development and improvement, from one-phase foam to two-phase foam materials, and then to multi-phase foam materials with the best performance and the widest application range [5]. Carbon fiber (CF), graphite, and glass fiber are all used as reinforced materials to enhance hollow spheres in the production of multi-phase ESF materials [6,7,8,9]. These reinforced materials have a relatively large density, which is not conducive to providing large buoyancy for deep-sea exploration equipment [10,11]. The introduction of the low-density aerogel is more conducive to reducing the density of foam to improve the overall performance and reduce the energy consumption of the submersible [12,13,14]. In this experiment, the aerogel reinforced hollow epoxy macro-spheres (AR-HEMS) was prepared innovatively by ‘the rolling ball method,’ and the same rolling time and method were adopted to make the aerogel powder wrapped around the EPS beads evenly [15,16]. This method can ensure that the surface of the macro-spheres are smooth, which is more conducive to the conduction of force inside the foam material.

In addition, a lot of research has been done on the reinforcement materials of the hollow macro-spheres [2,17], and the research on the EP-hardener-HGMS system is relatively few for the mulity-phase ESF. In order to further improve the compressive strength of deep-sea buoyancy materials, the introduction of a certain amount of CF material into the matrix is beneficial to improve the overall strength [18,19,20]. One-dimensional lightweight CF as a reinforced material can be closely combined with the EP-hardener-HGMS system. When the foam material is damaged by external force, the reinforced CF can effectively improve the integrity and bonding of the matrix [21]. For optimizing the compressive strength of the material, Wu et al. [22] increased the compressive strength of the ESF by about 7% by adding 300 W CF material into the epoxy resin and hollow galss microsphere (EP-HGMS) system. It can be seen that 300 W CF can increase the compressive strength of the ESF to a certain extent, but the effect is not obvious. Mana Halvaei et al. [23] used CF textiles with different mesh sizes and chopped CF to reinforce concrete materials. This experiment showed that the average strength of concrete materials increases as the length and content of chopped CF increase. When the CF is not added and the length of the CF is 4 mm and 8 mm, the average strength of the concrete composite material is 7.87 MPa, 12.15 MPa, and 15.35 MPa, respectively. Christoph Unterweger et al. [24] compounded 10 vol% CF materials with polypropylene materials. The tensile strength and tensile modulus of composite materials rise with the increase of fiber length. It can be seen that the length and content of CF have an important impact on the overall performance of the composite material to a large extent, but it is not that the longer the fiber length is, the more clear the improvement of the material performance is. Zenong Fang et al. compounded long fibers and short fibers with resin. Observed by SEM, CFs with shorter lengths are more evenly distributed in the composite material, and are more likely to exist in the holes and gaps in the material, which will be more beneficial to the overall performance.

In this experiment, the micron and millimeter level CF were used to study the influence of the length and content of CF on the compressive strength of the ESF. The influence of the volume stacking fraction, inner diameter, and layer of the AR-HEMS on the compressive strength of the ESF were also studied. From the data in the experiment, we found that, with the reinforced CF in the EP-hardener-HGMS, the density of the ESF is not changed clearly, but the compressive strength of the ESF is enhanced. Considering the two factors of the density and compressive strength, the ESF reinforced by 1.5 wt% CF with 90% AR-HEMS has better performance in the experiment. This kind of the ESF has 0.428 g/cm^3^ and 20.76 Mpa, which could be applied in 2076 m deep sea.

## 2. Materials and Methods

### 2.1. Materials

Epoxy resin Araldite LY 1564 is a white transparent liquid with a viscosity of about 40 mPa·s. Polyamine curing agent Aradur 3486 is a light-yellow liquid. The Brockfield viscosity and density of the resin are 2800 mPa·s and 1.17 g/cm^3^ at 25 °C, respectively. The color of the curing agent used for the experiment is dark amber. Both of the resin and the curing agent are manufactured by AXSON Technologies Co., Ltd., Shanghai, China.

The style of the hollow glass microspheres (HGMS) is S38HS, which is white. The density and compressive strength of the HGMS are 0.38 g/cm^3^ and 37.90 Mpa. This kind of the HGMS is manufactured by 3M company, MN, USA.

Aerogel is a nano-porous solid material formed by the sol-gel method, using a certain drying method to replace the liquid phase in the gel with gas. The main components of aerogel is SiO_2_. The specific surface area and bulk density of the aerogel are about 600 m^2^/g and 80 kg/m^3^. The porosity of the aerogel is about 90% to about 95%. This kind of the aerogel is manufactured by Zhongning Technology Co., Ltd., Shenzhen, China.

The EPS beads selected for the experiment are produced by Hangchao Packaging Materials Co., Ltd., Hangzhou, China. The density of the EPS beads is about 10 kg/m^3^.

The CF used for preparing the CF reinforced EP-HGMS (CFR-EP) and added into the EP-hardener-HGMS system is manufactured by Weida Composite Materials Co., Ltd., Nanjing, China. The density of the CF is about 1.76 g/cm^3^ and the wire diameter is about 7 μm.

### 2.2. Preparation of the AR-HEMS

First of all, three parts of epoxy resin and one part of curing agent were mixed, and the mixed solution was fully mixed under magnetic stirring. Next, a certain volume fraction of EPS beads was measured into a plastic container, and the prepared epoxy-hardener was poured slowly and evenly into the plastic container. The mixture was stirred by the glass rod so that the epoxy curing system can be completely applied to the surface of the EPS beads. The EPS beads covered by the epoxy-hardener system were poured into the running ball machine and enough aerogel powder was sprayed into the ball machine. When the aerogel powder was completely wrapped on the EPS beads, the excess aerogel powder on the surface of the beads was filtered and poured into the tray. After the epoxy resin on the surface of the EPS beads was completely cured, one layer of AR-HEMS has been prepared. By repeating the above steps once or twice, two-layer and three-layer AR-HEMS could be prepared. The schematic diagram of the preparation of the AR-HEMS is shown in Figure 1.

### 2.3. Preparation of the Multi-Phase ESF

Multi-phase ESF mainly includes AR-HEMS and CFR-EP system. First, three parts of epoxy resin and one part of curing agent were measured, respectively. The solution is fully mixed under the magnetic stirring, and then the HGMS with 150% of the epoxy-hardener volume fraction and CF were added into the epoxy-hardener system. At this time, the viscosity of the mixed slurry gradually increased, which has a certain degree of plasticity. After the HGMS, CF and the epoxy-hardener were completely mixed, an amount of AR-HEMS was added into the slurry, and the macro-spheres were evenly distributed in the CFR-EP system. The mixed slurry was slowly poured into the circular mold. The mold filled with the slurry was placed in a hot press and kept at 100 °C for 1 h, and then placed in a cold press for about 15 min. After the epoxy resin in the system was completely cured, the multi-phase ESF had been prepared. The schematic diagram of the preparation of the ESF is shown in Figure 2.

#### 2.3.1. Preparation of the ESF Filled with AR-HEMS with Different Stacking Volume Fractions

The influence of AR-HEMS with different stacking volume fractions on the compressive strength of ESF was studied in this group of experiments. Assuming that the volume fraction of the full mold is 100%, AR-HEMS with 20%, 40%, 60%, 80%, and 90% stacking volume fraction was added in the CFR-EP system with and without CF in this experiment. Table 1 shows 12 samples of the ESF filled with CFR-HEMS with different stacking volume fractions.

#### 2.3.2. Preparation of the ESF Filled with AR-HEMS with Different Layers

The influence of AR-HEMS with different layers on the compressive strength of ESF was studied in this group of experiments. AR-HEMS with no layer, 1 layer, 2 layers, and 3 layers were added into the CFR-EP system with and without CF in this experiment. Table 2 shows eight samples of ESF filled with CFR-HEMS with different wall thicknesses.

#### 2.3.3. Preparation of the ESF Filled with AR-HEMS with Different Inner Diameters

The influence of AR-HEMS with different inner diameters on the compressive strength of ESF was studied in this group of experiments. AR-HEMS with an inner diameter of 7–8 mm, 9–10 mm, and 11–12 mm were added into the CFR-EP system. Table 3 shows three samples of ESF filled with CFR-HEMS with different inner diameters.

#### 2.3.4. Preparation of the ESF Reinforced by Adding Different CF Content into an EP-Hardener-HGMS System

The influence of the CF content added into the EP-hardener-HGMS system on the compressive strength of ESF was studied in this group of experiments. 0%, 0.5%, 1%, 1.5%, 2%, and 2.5% CF were added into the EP-hardener-HGMS system in this experiment. Table 4 shows six samples of ESF prepared with EP-hardener-HGMS with different CF contents.

#### 2.3.5. Preparation of the ESF Reinforced by Adding Different Types of CF into EP-Hardener-HGMS System

The influence of the CF types added into the EP-hardener-HGMS system on the compressive strength of ESF was studied in this group of experiments. In total, 0.05 mm, 0.15 mm, 5 mm, and 10 mm of CF were added into the EP-HGMS system in this experiment. Table 5 shows five samples of ESF prepared with EP-hardener-HGMS reinforced by different CF types.

### 2.4. Characterization Methods

In order to measure the diameter and mass of the AR-HEMS, 50 samples of the AR-HEMS were chosen for measurement, which is beneficial to avoid the accidental data. A digital vernier caliper (SHAHE, KECHENG Instrument Co., Ltd., Ningbo, China) (Accuracy of 0.01 mm) and a digital analytical balance (FA1106, Biuged Laboratory Instruments Co. Ltd, Guangzhou, China) (Accuracy of 0.001 g) were used to measure the diameters and the mass of the AR-HEMS. The specification of the mould used to prepare the ESF is Ø70 mm × 65 mm, H60 mm, and the volumes of the samples are all 199 cm^3^.

The Compressive Properties of the ESF were tested by an electronic universal testing machine (CMT5350, Suns Technology Co., Ltd., Shenzhen, China). The GB/T16491-2008 is the standard in this experiment.

Scanning electron microscopy (SEM) (JEM-4701, JEOL, Tokyo, Japan) was used to observe the EP-HGMS and the AR-HEMS system. The observed sample should be thin enough and be sprayed by the gold.

## 3. Results

### 3.1. Physical Property of AR-HEMS

It can be seen in Figure 3 that five kinds of AR-HEMS with different particle sizes are all white spheres with regular shapes, and the surface of the sphere is smooth, which enhances the connection performance between the macro-sphere and the CFR-EP system. The AR-HEMS with different inner diameters and different layers are showed in Figure 3a,b. As can be seen in Figure 4b, the diameters of AR-HEMS are divided into three areas: (a) 9.01–9.83 mm, (b) 9.14–10.51 mm, and (c) 9.39–10.98 mm. Since the diameter of the AR-HEMS changes within 1 mm, the diameters of AR-HEMS with different layers still have most of the overlap area. The average volume of the AR-HEMS increases as the AR-HEMS layer increases, but its increasing trend is relatively gentle. The mass of the sphere increases significantly due to the increase in the content of the coating resin and aerogel. The average density of the AR-HEMS can be calculated by Equation (1).

This is the first example of an equation:(1)$$\rho_{AR-HEMS}=\;\frac{\sum_{n=0}^{n=60}{\left(m_{n}\right)}_{AR-HEMS}}{\sum_{n=0}^{n=60}{\left(V_{n}\right)}_{AR-HEMS}}$$

As can be seen from Figure 4b, the density of AR-HEMS spheres with 1, 2, and 3 layers are 0.056 g/cm^3^, 0.143 g/cm^3^, and 0.198 g/cm^3^. Although the strength of the sphere increases with the rise in the layers, the growth in the density of AR-HEMS will also lead to an increase in the density of the ESF material. In the deep-sea field, the increase in the quality will inevitably lead to the need for more power to drive the equipment to work. The great mass also increases the equipment load and reduces the service life to a certain extent. Therefore, when preparing ESF, it is necessary to select the appropriate layer of AR-HEMS, according to the specific working environment.

Figure 4a shows AR-HEMS with inner diameters of 7–8 mm, 9–10 mm, and 11–12 mm. The sphere is smooth and regular. It can be clearly seen that the diameter of AR-HEMS increases with the growth of the inner diameter. The diameter of AR-HEMS is roughly divided into three areas: (a) 7.19–8.69 mm, (b) 9.14–10.51 mm, and (c) 11.05–12.41 mm. It can be seen from Figure 4a that the density of AR-HEMS decreases as the inner diameter increases. The density of the three kinds of the AR-HEMS are 0.183 g/cm^3^, 0.143 g/cm^3^, and 0.142 g/cm^3^, respectively. When the layer is the same, the larger the inner diameter of AR-HEMS is, the smaller the strength of the sphere is, but its density also decreases. Therefore, density and strength are two contradictory materials. It is difficult to achieve both two good performances for the ESF materials. The key to preparing buoyant materials is to balance the density and compressive strength of ESF materials.

### 3.2. Physical Property of ESF

Figure 5 shows the cross section of ESF material. From the figure, it can be seen that ESF material is mainly divided into the two parts of the CFR-EP and AR-HEMS system. The thickness of the round, white, spherical wall is uniform and the wall is smooth. EPS beads were shrunken in the wall of the AR-HEMS for the high temperature. The diameter of the core is about 1/3~1/4 that of the AR-HEMS. Therefore, the AR-HEMS is hollow. The white wall of the AR-HEMS and the CFR-EP are the same resin system. It can be seen from Figure 5 that the two parts are connected tightly and have almost no gap between them, which improve the overall performance of the ESF. The color of the EP-HGMS is white originally. The color becomes black gradually because of the addition of the CF. From Figure 6, we could see that, besides the hollow AR-HEMS, there are some small bubbles in the surface of the ESF. The generation of the bubbles is mainly because the added HGMS and CF make the viscosity of the matrix higher. Some bubbles are inevitable to be generated during the stirring and pouring. These bubbles will reduce the compressive strength of the ESF, but the existence of the bubbles also reduce the overall density. In all, the internal integrity and the combination of different systems are good, which also provide relevant suggestions for the preparation and application of ESF materials.

### 3.3. Influence of Stacking Volume Fraction of AR-HEMS on the Compressive Property of ESF

In order to study the influence of AR-HEMS with different stacking volume fractions on the compressive strength of ESF, 20%, 40%, 60%, 80%, and 90% AR-HEMS were added into the matrix in this group. As shown in Figure 7a, the compressive strength of ESF decreases as the stacking volume fraction increases. When the stacking volume fraction of the macro-spheres is 20%, 40%, 60%, 80%, and 90%, the compressive strength of the ESF are 32.05 Mpa, 26.71 Mpa, 22.05 Mpa, 20.01 Mpa, and 19.46 Mpa. From these five data points, we can get that, although the compressive strength of ESF is decreasing, its decrease tends to be gentle. When the stacking volume fraction is 60%, 80%, and 90%, the decrease trend of the compressive strength is much smaller than the ESF with 20% and 40% AR-HEMS. There are some reasons for this phenomenon. (a) As shown in Figure 4b, because the density of AR-HEMS is less than the density of the matrix, the hollow spheres will float up during the material forming process. AR-HEMS will appear in the upper middle area of the ESF. This floating phenomenon leads to uneven distribution inside the material. This phenomenon is more clear when the stacking volume fraction is low. When the content of AR-HEMS increases, the volume fraction of ESF material occupied by AR-HEMS gradually increases, which greatly alleviates the floating phenomenon. The distribution in the composite material tends to be uniform, which is the main reason why the decreasing trend of the compression strength is getting slower. (b) When the stacking volume fraction of AR-HEMS is relatively low, the AR-HEMS is more scattered and the chance of contact is lower. As the stacking volume fraction of AR-HEMS increases, as shown in Figure 8, the AR-HEMS are in contact with each other to form a relatively complete pressure-bearing network, which provides conditions for the force transmission inside the composite material.

Although the compression strength of ESF decreases with the increase of the stacking volume fraction, the density of the composite material decreases. As shown in Figure 7b, when the stacking volume fraction of the macro-spheres are 20%, 40%, 60%, 80%, and 90%. The density of ESF are 0.65 g/cm^3^, 0.60 g/cm^3^, 0.53 g/cm^3^, 0.49 g/cm^3^, 0.45 g/cm^3^, and 0.423 g/cm^3^. When foam buoyancy materials are used in subsea equipment, ESF must have higher compressive strength to resist deep sea pressure, but lightweight materials can greatly reduce the energy consumption and provide more buoyancy. Therefore, strength and density are both important factors when preparing deep-sea buoyancy materials. In this group of experiments, when the stacking volume fraction is 90%, the ESF has the lower density and better compressive strength relatively, which can be used for subsea equipment working at a depth of about 1900 m.

### 3.4. Influence of the Layers of AR-HEMS on the Compressive Property of ESF

In order to study the effect of the layers of AR-HEMS on the compression strength of ESF, 0-layer, 1-layer, 2-layer, and 3-layer AR-HEMS were added to the matrix with the stacking volume fraction of 90% in this experiment. As shown in Figure 9, when the layer is 0-layered, the EPS is added into the matrix without any reinforcement. There are a lot of holes on the surface of the ESF. These holes are due to the shrinkage of the EPS beads under high temperature, which has a huge impact on the compressive strength of ESF. When the EPS beads without any reinforcement are placed in the matrix, although the density of the composite material is reduced (0.36 g/cm^3^), it destroys the integrity of the ESF material. Therefore, the 0-layer AR-HEMS is not suitable for the preparation of the ESF materials. As shown in Figure 10a, when the layer of the AR-HEMS is 0, 1, 2, and 3, the compressive strength of the composite material is 10.14 MPa, 14.26 MPa, 19.46 MPa, and 23.15 MPa. From this set of data, when the AR-HEMS is 1 layer, the ESF material has a lower density (0.38 g/cm^3^), but it is limited by the lower compressive strength, which is about 14.26 Mpa. In contrast, when the AR-HEMS is three layers, the ESF material has the higher density (0.45g/cm^3^), but this type of ESF has a wider range of applications and can be used for subsea equipment in the 2315-m deep sea. Therefore, the selection of materials during manufacturing must be based on the working environment and work requirements and the two performances of the density and compressive strength need to be weighed.

### 3.5. Influence of the Inner Diameters of AR-HEMS on the Compressive Property

In order to study the influence of a different inner diameter of AR-HEMS on the compressive strength of ESF, in this group of experiment, AR-HEMS with 7–8 mm, 9–10 mm, and 11–12 mm inner diameter were selected as fillers, and added to the matrix. As shown in Figure 11a, the ESF material with AR-HEMS of a 7–8-mm inner diameter has the largest compressive strength, which can reach 24.24 Mpa. When the inner diameter of the hollow sphere is smaller, it can withstand a greater load, which also enhances the compressive strength of the composite material to a certain extent. When the inner diameter of AR-HEMS is 11–12 mm, the compressive strength of ESF is greatly reduced with only 67.7% (16.41Mpa) of the former. For equipment and materials used in the deep-sea field, compressive strength and density are always a contradiction. The two need to be balanced to make the use of composite materials wider. Although the ESF prepared by AR-HEMS with the inner diameter of 7–8 mm has a large compressive strength. Its density is also relatively large, which is about 0.46 g/cm^3^. The ESF prepared by AR-HEMS with an inner diameter of 11–12 mm has a lower density, which is about 0.41 g/cm^3^, but its compressive strength is not high. In this group of experiments, ESF material prepared by the AR-HEMS with the inner diameter of 9–10 mm has relatively high compressive strength and can also provide more buoyancy for deep-sea equipment. Therefore, AR-HEMS with an inner diameter of 9–10 mm is the best choice for preparing buoyant materials in this experiment.

### 3.6. Influence of the Content of CF in CFR-EP on the Compressive Property of ESF

In order to study the influence of different content of CF added into the matrix on the compressive strength of the ESF, 5-mm CF was selected in this group of experiments. As shown in Figure 12a, the compressive strengths are 15.75 Mpa, 17.74 Mpa, 18.91 Mpa, 19.46 Mpa, 22.07 Mpa, and 21.09 Mpa after adding 0%, 0.5%, 1%, 1.5%, 2%, and 2.5% of CF to the matrix, respectively. However, the density of ESF does not change much, which is changing between 0.41 g/cm^3^ to 0.43 g/cm^3^. It can be seen from the data that the compressive strength of ESF increases with the increase of CF content. When the CF content is 1.5%, the CF is evenly distributed in the matrix, so the compressive strength of ESF material reaches a maximum value. It can be seen in the SEM of the matrix that the fiber and resin are tightly combined. When a load is applied to the ESF material from the outside, the CF will enhance the bonding force of the resin matrix due to its good tensile strength, and enhance the compressive strength of the composite material.

As shown in Figure 12a, when the CF content reaches 2.5%, the compressive strength of the ESF material changes and shows a downward trend. There are several reasons for this phenomenon: (a) The excessive addition of CF leads to an increase in the possibility of agglomeration in the matrix, which reduces the interface bonding force between the resin and the CF, and more defects are generated in the composite material. (b) The addition of excessive CF leads to an increase of the viscosity of the resin matrix, which also increases the difficulty of preparing the composite material, and produces certain hole defects during the processing and molding process. Therefore, the compressive strength of the ESF cannot be enhanced by increasing the content of CF alone.

### 3.7. Influence of the Types of CF in EP-HGMS on the Compressive Property of ESF

In order to study the effect of different lengths of CFs on the compression strength of ESF, 0.05 mm, 0.15 mm, 2 mm, and 5 mm of CFs were chosen in this experiment. 1.5% mass fraction of CF were added into the matrix. As shown in Figure 13b, the density of the ESF has not changed much and remains at about 0.43 g/cm^3^. As shown in Figure 13a, when the CF is at the micron level, the compressive strength of ESF are 17.93 Mpa (0.05 mm) and 18.71 Mpa (0.15 mm). When the CF is at the millimeter level, the compressive strength of ESF are 19.46 Mpa (2 mm) and 20.76 Mpa (5 mm). From this set of data, as the length of the CF increases, the compressive strength of the ESF is also improved. The addition of millimeter-level CFs is more effective in improving the compressive strength of ESF than the micron level. The compression strength of 5-mm CF reinforced ESF is about 15.8% higher than that of 0.05-mm CF reinforced ESF. The main reason for this phenomenon is that CF with a mass fraction of 1.5% can be evenly distributed in the resin matrix, and longer CF can be more closely combined with the resin matrix, which further improves the integrity of the composite material. When the ESF is under load, longer CF can provide the ESF with greater compression resistance.

### 3.8. Influence of the Addition of CF in EP-HGMS on the Compressive Property of ESF

As a lightweight material with extremely strong tensile strength, CF is always added to the polymer to improve the overall performance of the composite material, which has become the mainstream of making high-strength and low-density materials. In this experiment, CF is mixed with resin matrix and HGMS. From the SEM picture of the composite material, it can be seen that the CF is distributed evenly in the matrix, and there is no CF agglomeration. The CF reinforced resin material, which is uniform, is extremely conducive to the conduction of the load inside the composite material. The CF and the resin matrix are tightly combined without clear material defects. The addition of CF has no major effect on the density of the composite material. However, the compressive strength of the composite material has been significantly improved with the addition of CF. It could be found that the compressive strength of the ESF prepared by the AR-HEMS with different stacking volumes without reinforeced CF in the EP-hardener-HGMS system was showed in Figure 14a. The enhancement of ESF compressive strength by 1.5% CF are obviously exhibited in Figure 14b.

The addition of CF increases the viscosity of the resin system, which prevents the floating of the EPS beads to a certain extent. When adding unreinforced EPS beads to the resin matrix, if CF is not added to the resin matrix, the ESF material will not be able to form. A large number of EPS beads was poured into the upper half from the lower half during the forming process. Thin shell-like materials are easily damaged during demolding, and, finally, appear as a composite material collapsed in the upper half, as shown in Figure 15. This defective material cannot meet production requirements. When the unreinforced EPS are added to the CF-reinforced resin matrix, although the material can be molded and maintain a certain strength, its surface has many holes, which still does not meet the production requirements. Compared with the ESF without the CF being reinforced, the resin could be stickier, which prevent the floating of the AR-HEMS. The compressive strength of the ESF prepared by the AR-HEMS with different layers without reinforeced CF in the EP-hardener-HGMS system was shown in the Figure 16a. It could be noticed that when AR-HEMS is 1, 2, and 3 layers, the enhancement of ESF compressive strength by 1.5% CF are 80% (1 layer) and 23.6% (2 layers) and 25.3% (3 layers) in Figure 16b. As is shown in Figure 14, the EP-hardener-HGMS system is also reinforced by the CF, which enhance the compressive strength of the ESF. When AR-HEMS is 1, 2, and 3 layers, 1.5% CF is added to the matrix, as shown in Figure 16b, the enhancement of ESF compression strength by CF is 80% (1 layer) and 23.6% (2 layers) and 25.3% (3 layers).

### 3.9. SEM Image of the ESF

Figure 17a shows the brittle section near the AR-HEMS wall. The material is mainly divided into three parts: the CFR-EP system, the wall of AR-HEMS, and the inner wall. It can be seen from Figure 17a that, since the spherical wall and the matrix are both epoxy systems, the two parts are closely combined. Figure 17b is an enlarged part from the blue circle in Figure 17a. Figure 17b clearly shows that there are almost no defects in the connection between the two systems, which provides favorable conditions for the conduction of force inside the material. The load distributed inside the material is more even, which protects the ESF from breaking and improves the compressive strength.

Figure 18 shows the picture of the Energy Dispersive Spectrometer (EDS) near the spherical wall. It can be seen from the figure that the spherical wall of the AR-HEMS system composed of aerogel and epoxy resin are evenly separated and tightly bonded, which is beneficial to enhance the compressive strength of the material. The silicon is the main components of aerogel and HGMS. It can be seen from Figure 18 that the distribution of silicon in the material is relatively uniform, and it mainly exists in the spherical wall of AR-HEMS and CFR-EP system. The two systems are more distinct in the EDS picture and have good combination, which creates conditions for the preparation of high-performance ESF material.

Figure 17c is a SEM picture of the CF enhanced EP-hardener-HGMS system. The materials mainly include white spherical HGMS and one-dimensional CF materials. It can be seen from Figure 18 that the HGMS have no clustering phenomenon and are evenly distributed, and the CF material is extremely uniform. The two fillers are firmly combined with the epoxy resin matrix, and there are no holes in the body, which provides favorable conditions for improving the overall mechanical properties. The addition of lightweight CF can enhance the compressive strength to a certain extent under the condition that the density of the composite material is not significantly affected. When the material is loaded by the outside force, CF can prevent the damage, which also provides certain suggestions and methods for the preparation of the buoyance material.

## 4. Conclusions

In all, six compared groups were carried out in this experiment. The ESF prepared in this experiment includes two parts of AR-HEMS and CFR-EP system. From the test of the performance and appearance of the ESF, some conclusions could be made.

(a) The AR-HEMS prepared in this experiment is almost round, which indicates that the ‘rolling ball method’ is beneficial for the preparation of the AR-HEMS. The regular sphere added into the matrix is conducive to combine with the matrix, which enhances the compressive strength of the ESF. From the data of the AR-HEMS, we could see that the density of the AR-HEMS increases with the layer increasing and the inner diameter decreasing.

(b) The compressive strength of ESF decreases as the stacking volume fraction increases, and its decrease tends to be gentle. However, the density of the composite material decreases. When the stacking volume fraction is 90%, the ESF has the lower density (0.423 g/cm^3^) and better compressive strength (19.46 Mpa) relatively. The compressive strength and density are both important factors when preparing deep-sea buoyancy materials.

(c) The compressive strength of ESF increases as the layer grows, but its density gets larger. Therefore, the selection of the layer during manufacturing must be based on the working environment and work requirements of different equipment.

(d) For equipment and materials used in the deep-sea field, compressive strength and density are always a contradiction. The compressive strength of ESF increases as the inner diameter decreases, but its density gets larger. The ESF prepared with AR-HEMS with the inner diameter of 9-10 mm has relatively better performance.

(e) When the CF content is from 0%~2%, the compressive strength of ESF increases with the increase of CF content. However, when the CF content is more, the compressive strength of the ESF material changes and shows a downward trend because of the material defects.

(f) 1.5% CF can be evenly distributed in the resin matrix, and longer CFs can be more closely combined with the resin matrix, which further improves the integrity of the composite material.

(g) When the AR-HEMS accumulation ratio is 20%, the compressive strength of the ESF reinforced by CF material has the biggest change, which is increased by about 28% compared with the ESF without CF. The addition of the CF could increase the viscosity of the matrix, which prevents the float of the AR-HEMS. The CF as the reinforced material in the matrix could also improve the strength of the EP-hardener-HGMS system.

## Figures and Tables

**Figure 1 polymers-13-00683-f001:**
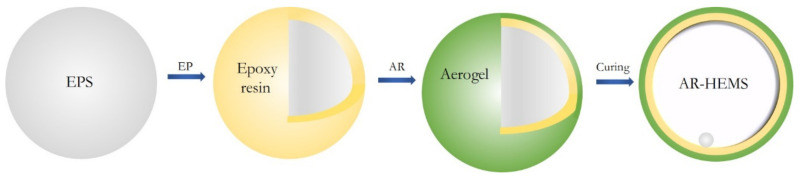
The schematic diagram of the preparation of the AR-HEMS.

**Figure 2 polymers-13-00683-f002:**
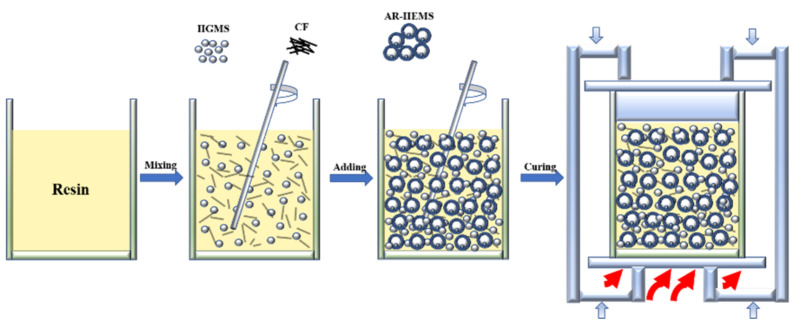
The schematic diagram of the preparation of the ESF.

**Figure 3 polymers-13-00683-f003:**
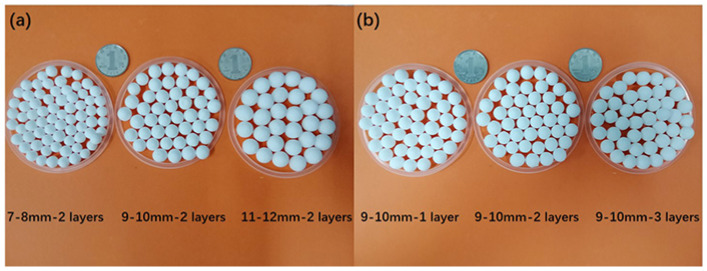
AR-HEMS with a different inner diameter and layer.

**Figure 4 polymers-13-00683-f004:**
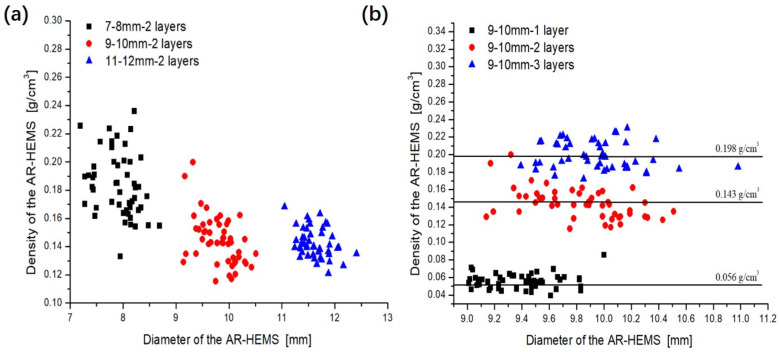
Scatter plot of the relationship between the diameter and density: (**a**) inner diameter is 9–10 mm and (**b**) the layer is 2-layered.

**Figure 5 polymers-13-00683-f005:**
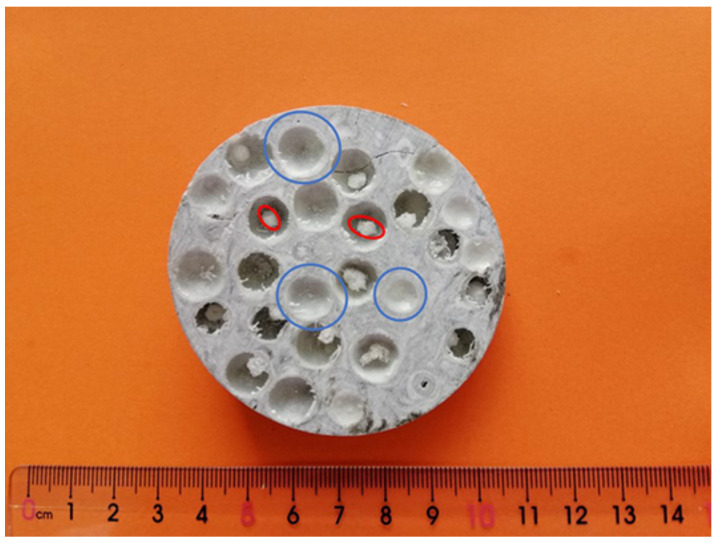
The cross section of the ESF.

**Figure 6 polymers-13-00683-f006:**
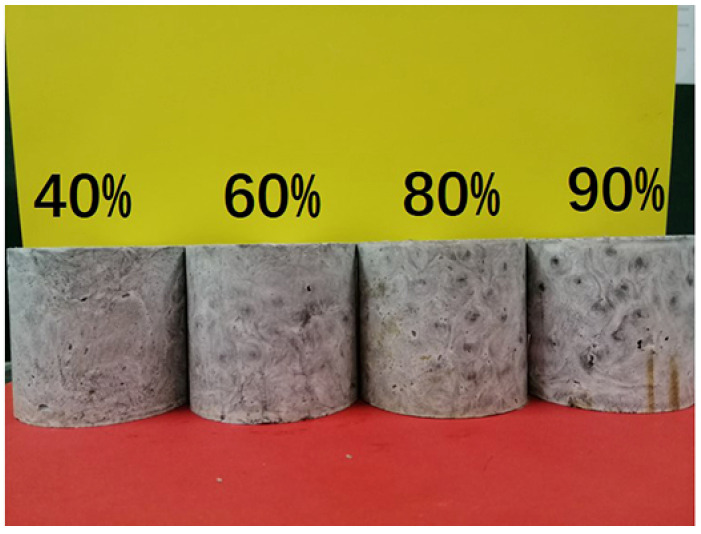
ESF filled with AR-HEMS with a different stacking volume fraction.

**Figure 7 polymers-13-00683-f007:**
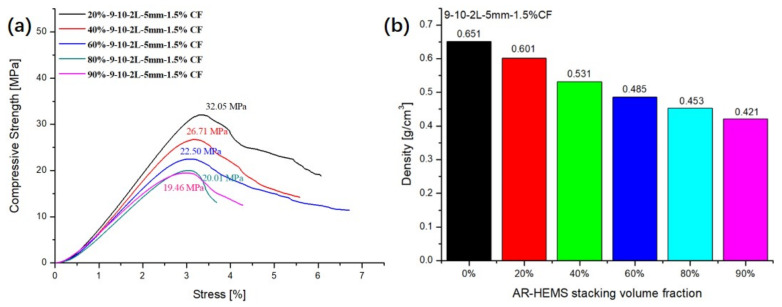
Influence of the stacking volume fraction: (**a**) compressive strength and (**b**) density.

**Figure 8 polymers-13-00683-f008:**
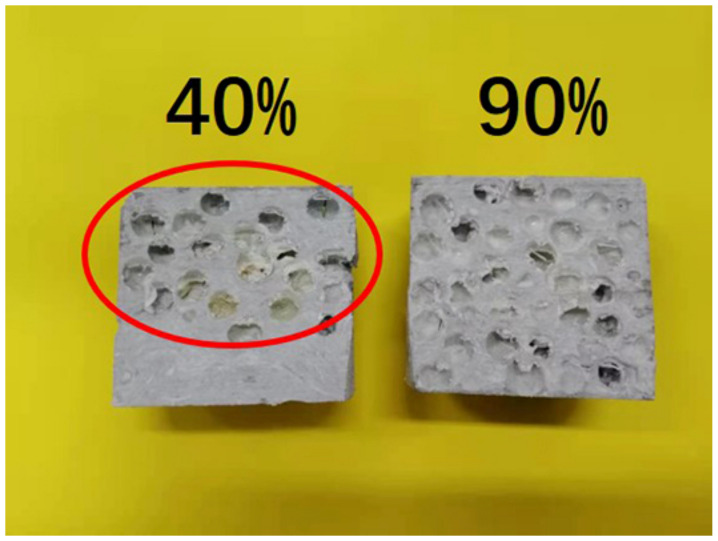
The longitudinal section of the ESF.

**Figure 9 polymers-13-00683-f009:**
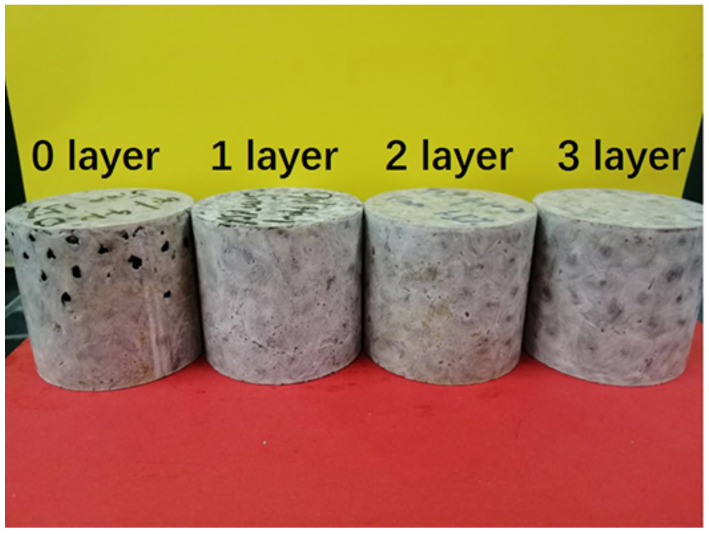
ESF filled with 90% AR-HEMS with different layers.

**Figure 10 polymers-13-00683-f010:**
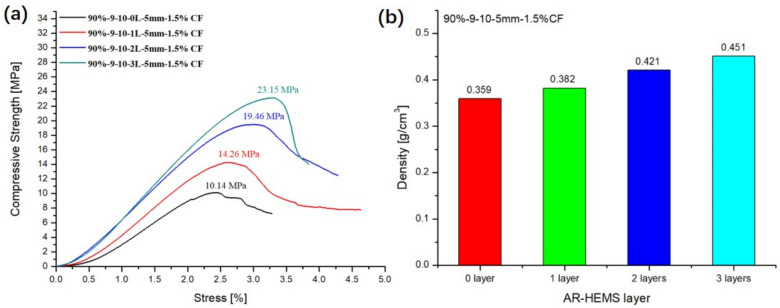
Influence of the layers of AR-HEMS: (**a**) compressive strength and (**b**) density.

**Figure 11 polymers-13-00683-f011:**
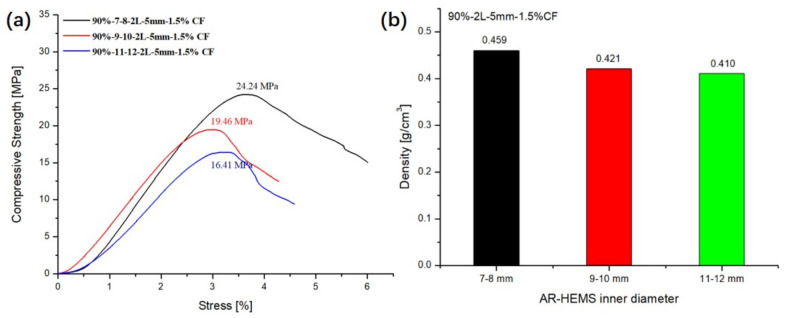
Influence of the inner diameters of AR-HEMS: (**a**) compressive strength and (**b**) density.

**Figure 12 polymers-13-00683-f012:**
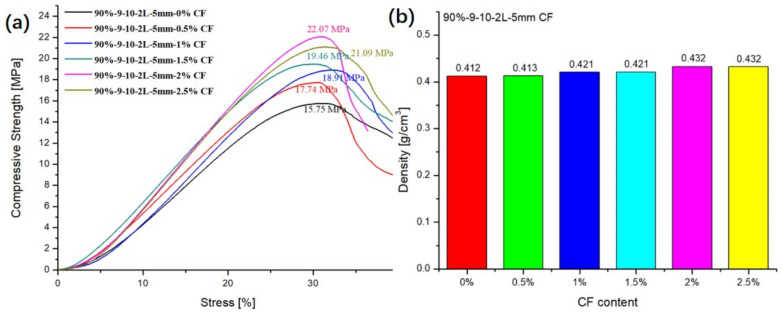
Influence of the content of CF in CFR-EP: (**a**) compressive strength and (**b**) density.

**Figure 13 polymers-13-00683-f013:**
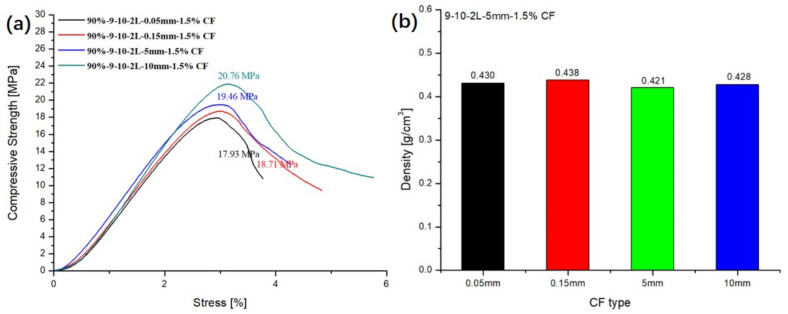
Influence of the length of CF in CFR-EP: (**a**) compressive strength and (**b**) density.

**Figure 14 polymers-13-00683-f014:**
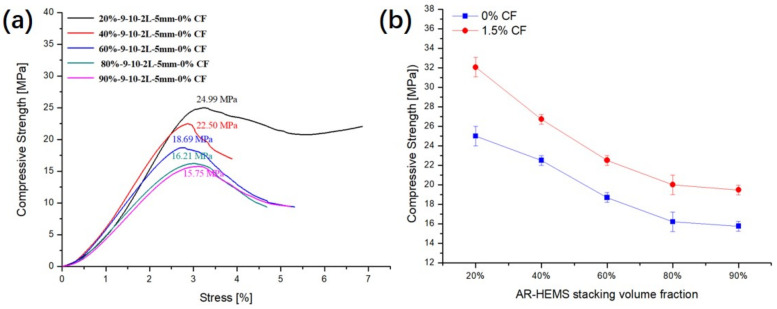
Influence of the addition of CF in EP-HGMS on the compressive properties of ESF with AR-HEMS with different stacking volume fractions.

**Figure 15 polymers-13-00683-f015:**
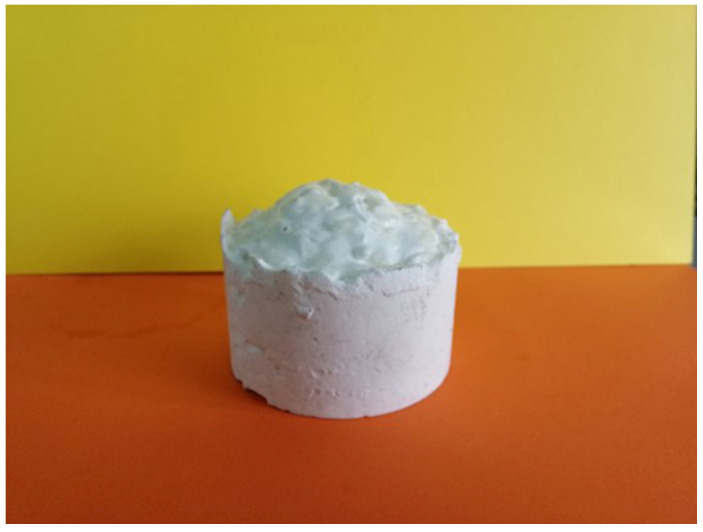
The unshaped ESF.

**Figure 16 polymers-13-00683-f016:**
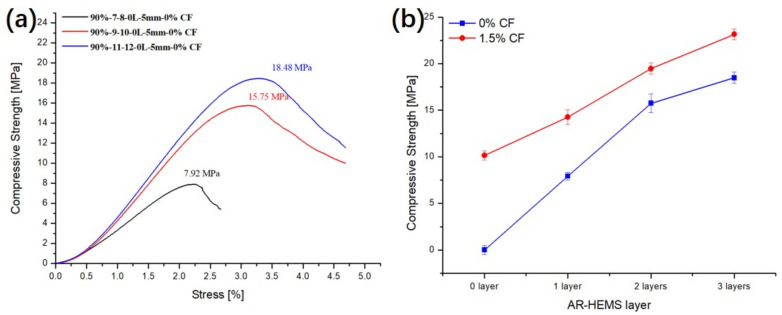
Influence of the addition of CF in EP-HGMS on the compressive properties of ESF with AR-HEMS with different layers.

**Figure 17 polymers-13-00683-f017:**
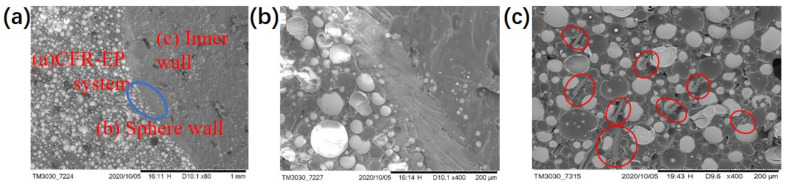
SEM image of the sphere wall and matrix.

**Figure 18 polymers-13-00683-f018:**
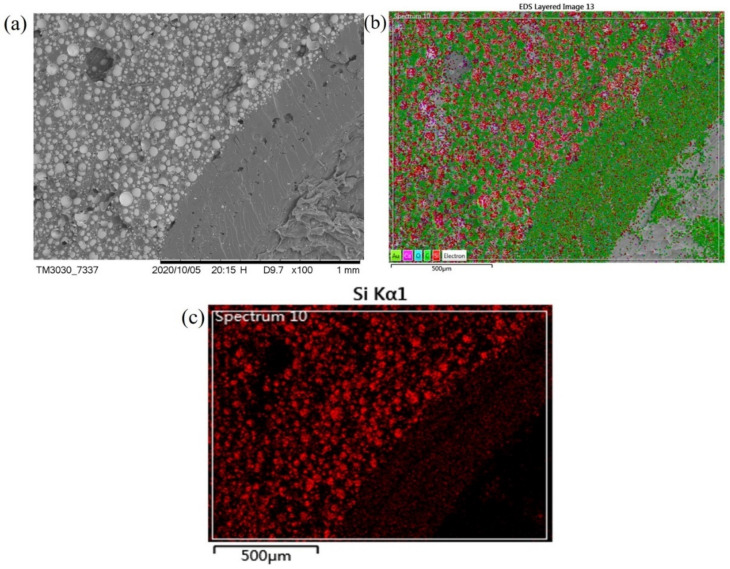
(**a**) SEM image and (**b**, **c**) EDS of the sphere wall.

**Table 1 polymers-13-00683-t001:** 12 samples of ESF filled with AR-HEMS with different stacking volume fractions.

Stacking Volume Fraction of AR-HEMS	Wall Thickness of AR-HEMS	Inner Diameters of AR-HEMS	HGMS Content	Resin Type	CF Type in HGMS-EP	CF content in HGMS-EP
0%	2 layers	9–10 mm	60%	1564/3486	5 mm	1.5%/0%
20%	2 layers	9–10 mm	60%	1564/3486	5 mm	1.5%/0%
40%	2 layers	9–10 mm	60%	1564/3486	5 mm	1.5%/0%
60%	2 layers	9–10 mm	60%	1564/3486	5 mm	1.5%/0%
80%	2 layers	9–10 mm	60%	1564/3486	5 mm	1.5%/0%
90%	2 layers	9–10 mm	60%	1564/3486	5 mm	1.5%/0%

**Table 2 polymers-13-00683-t002:** Eight samples of ESF filled with AR-HEMS with different layers.

Stacking Volume Fraction of AR-HEMS	Wall Thickness of AR-HEMS	Inner Diameters of AR-HEMS	HGMS Content	Resin Type	CF Type in HGMS-EP	CF Content in HGMS-EP
90%	0 layer	9–10 mm	60%	1564/3486	5 mm	1.5%/0%
90%	1 layer	9–10 mm	60%	1564/3486	5 mm	1.5%/0%
90%	2 layers	9–10 mm	60%	1564/3486	5 mm	1.5%/0%
90%	3 layers	9–10 mm	60%	1564/3486	5 mm	1.5%/0%

**Table 3 polymers-13-00683-t003:** Three samples of ESF filled with AR-HEMS with different inner diameters.

Stacking Volume Fraction of AR-HEMS	Wall Thickness of AR-HEMS	Inner Diameters of AR-HEMS	HGMS Content	Resin Type	CF Type in HGMS-EP	CF Content in HGMS-EP
90%	2 layers	7–8 mm	60%	1564/3486	5 mm	1.5%
90%	2 layers	9–10 mm	60%	1564/3486	5 mm	1.5%
90%	2 layers	11–12 mm	60%	1564/3486	5 mm	1.5%

**Table 4 polymers-13-00683-t004:** Six samples of ESF reinforced by CF with different contents.

Stacking Volume Fraction of AR-HEMS	Wall Thickness of AR-HEMS	Inner Diameters of AR-HEMS	HGMS Content	Resin Type	CF Type in HGMS-EP	CF Content in HGMS-EP
90%	2 layers	9–10 mm	60%	1564/3486	5 mm	0%
90%	2 layers	9–10 mm	60%	1564/3486	5 mm	0.5%
90%	2 layers	9–10 mm	60%	1564/3486	5 mm	1%
90%	2 layers	9–10 mm	60%	1564/3486	5 mm	1.5%
90%	2 layers	9–10 mm	60%	1564/3486	5 mm	2%
90%	2 layers	9–10 mm	60%	1564/3486	5 mm	2.5%

**Table 5 polymers-13-00683-t005:** Five samples of ESF reinforced by CF with different lengths.

Stacking Volume Fraction of AR-HEMS	Wall Thickness of AR-HEMS	Inner Diameters of AR-HEMS	HGMS Content	Resin Type	CF Type in HGMS-EP	CF Content in HGMS-EP
90%	2 layers	9–10 mm	60%	1564/3486	0	1.5%
90%	2 layers	9–10 mm	60%	1564/3486	0.05 mm	1.5%
90%	2 layers	9–10 mm	60%	1564/3486	0.15 mm	1.5%
90%	2 layers	9–10 mm	60%	1564/3486	5 mm	1.5%
90%	2 layers	9–10 mm	60%	1564/3486	10 mm	1.5%

## Data Availability

The data presented in this study are available on request.

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
