# Peer review of "Carbon Fiber Reinforced Multi-Phase Epoxy Syntactic Foam (CFR-Epoxy-Hardener/HGMS/Aerogel-R-Hollow Epoxy Macrosphere(AR-HEMS))"

_polymers, 2021, doi:10.3390/polym13050683_

Round 1

Reviewer 1 Report

The paper is interesting and well structured. I think it can be accepted in the present form.

Author Response

Response to Reviewer 1

Point 1: The paper is interesting and well structured. I think it can be accepted in the present form.

Response 1: Thank you for your suggestions.

Reviewer 2 Report

Comments to the authors: This is very interesting work and an obvious testing and characterizations have been done; however, some minor revisions need to be done to increase its value. - In material section, resin name should be presented completely instead of the code; “1564/3486 resin” is not enough. - Also, in addition to hardener’s name “S38HS”, its full name/group name should be presented. - In addition to the aerogel specifications, its scientific/commercial name should be stated. - Line 115: “3 parts and 1 part of epoxy resin and curing agent were mixed respectively”, should be “3 parts of epoxy resin and 1 part of curing agent were mixed”, - Line 118: “The mixture is stirred by the glass rod” should be “The mixture was stirred by …”. - Line 121 & 134 : what is the “certain amount”? This needs to be quantified clearly. - Line 124: “… the EPS beads is completely cured” should be “… the EPS beads was completely cured” - Line 137: “ a amount of” ? - Text size of all graphs as well as curves’ line thickness starting form Figure 4, need to be increased and adjust to present them properly. - In conclusion, main achievement should be presented quantitively. Thank you. Best Regards, Reviewer

Author Response

Response to Reviewer 2

Thank you for your suggestions very much. I have read the content seriously and made some revises below.

Point 1: In material section, resin name should be presented completely instead of the code; “1564/3486 resin” is not enough. - Also, in addition to hardener’s name “S38HS”, its full name/group name should be presented. In addition to the aerogel specifications, its scientific/commercial name should be stated.

Response 1: Thank you for your suggestions very much. The related introduction about the 1564 epoxy resin, 3486 curing, S38HS and aerogel has been described in the paper. Epoxy resin Araldite LY 1564 is a white transparent liquid with a viscosity of about 40 mPa·s. Polyamine curing agent Aradur 3486 is a light yellow liquid. The style of the hollow glass microspheres (HGMS) is S38HS, which is white. The density and compressive strength of the HGMS are 0.38 g/cm3 and 37.90 Mpa. Aerogel is a nano-porous solid material formed by sol-gel method, using a certain drying method to replace the liquid phase in the gel with gas. The main components of aerogel is SiO2.

Point 2: Line 115: “3 parts and 1 part of epoxy resin and curing agent were mixed respectively”, should be “3 parts of epoxy resin and 1 part of curing agent were mixed”. Line 118: “The mixture is stirred by the glass rod” should be “The mixture was stirred by …”.

Response 2: Thank you for your suggestions very much. The sentence has been replaced in the paper.

Point 3: Line 121 & 134 : what is the “certain amount”? This needs to be quantified clearly.

Response 3: Thank you for your suggestions very much. The aerogel was added enough to make the HEMS wrapped and then the excess powder was sifted away. The HGMS with 150% of the epoxy-hardener volume fraction and CF were added into the epoxy-hardener system.

Point 4: Line 124: “… the EPS beads is completely cured” should be “… the EPS beads was completely cured”

Response 4: Thank you for your suggestions very much. The sentence has been replaced in the paper.

Point 5: Line 137: “ a amount of” ?

Response 5: Thank you for your suggestions very much. The specific content has been explained in the section 2.3.1.

Point 6: Text size of all graphs as well as curves’ line thickness starting form Figure 4, need to be increased and adjust to present them properly.

Response 6: Thank you for your suggestions very much. Text size of all graphs as well as curves’ line thickness starting form Figure 4 have been corrected.

Reviewer 3 Report

The manuscript "Carbon fiber reinforced multi-phase epoxy syntactic foam  2 (CFR-epoxy-hardener/HGMS/Aerogel-R-hollow epoxy macrosphere(AR-HEMS))" has as main objective manufacture a composite susing epoxy resin-hardner system, aerogel and carbon fiber for the purpose of producing a floating material. The study presents the quantitative results obtained from experimental tests of density, volume fraction and compressive strenght, and SEM of each group of samples. They are compared and the conclusions are made.

The manuscript is original and has its merit, but it needs to be improved.

Some informatiosn on methodology must be improved.

Correct the formatting mistakes on units and references citations.

Review writing english.

The figures size and resolution must be improved.

All items to be corrected were highlighted in yellow in pdf document.

Author Response

Response to Reviewer 3

Point 1:The manuscript "Carbon fiber reinforced multi-phase epoxy syntactic foam  2 (CFR-epoxy-hardener/HGMS/Aerogel-R-hollow epoxy macrosphere(AR-HEMS))" has as main objective manufacture a composite susing epoxy resin-hardner system, aerogel and carbon fiber for the purpose of producing a floating material. The study presents the quantitative results obtained from experimental tests of density, volume fraction and compressive strenght, and SEM of each group of samples. They are compared and the conclusions are made.

Response 1: Thank you for your suggestions.

Point 2: Correct the formatting mistakes on units and references citations. Review writing english. The figures size and resolution must be improved. All items to be corrected were highlighted in yellow in pdf document.

Response 2: Thank you for your suggestions. The formatting mistakes on units and references citations have been corrected. The figures size and resolution have been improved. All items have been highlighted in yellow in paper.
